# The Effect of Knocked-Down Anti-Müllerian Hormone mRNA on Reproductive Characters of Male Nile Tilapia (*Oreochromis niloticus*) through Inhibition of the TGF-Beta Signaling Pathway

Yue Yan [1], Yifan Tao [2,*], Zheming Cao [2], Siqi Lu [2], Pao Xu [2] and Jun Qiang [1,2,*]

[1] Wuxi Fisheries College, Nanjing Agricultural University, Wuxi 214081, China
[2] Key Laboratory of Freshwater Fishes and Germplasm Resources Utilization, Ministry of Agriculture, Freshwater Fisheries Research Center, Chinese Academy of Fishery Sciences, Wuxi 214081, China
* Correspondence: taoyifan@ffrc.cn (Y.T.); qiangj@ffrc.cn (J.Q.)

**Abstract:** Anti-Müllerian hormone (*amh*), an important regulator of gonad development in male teleosts, regulates the development and differentiation of germ cells. We performed transcriptional knock-down of *amh* in Nile tilapia (*Oreochromis niloticus*) using antisense RNA technology, resulting in down-regulation in the expression of *amh* transcription and Amh protein in males. Compared with the control groups, the fish in treatment groups with down-regulated *amh* had increased weight and an extremely significant decrease in the gonadosomatic index. Hematoxylin–eosin staining revealed impaired testis development and significant reductions in numbers of sperm. Serum estradiol levels were significantly increased, and the levels of testosterone, luteinizing hormone, and follicle-stimulating hormone were significantly decreased. RNA-sequencing analysis of the fish in the down-regulated *amh* and control groups identified 12,048 differentially expressed genes, of which 1281 were up-regulated and 10,767 were down-regulated. Kyoto Encyclopedia of Genes and Genomes analysis revealed that differentially expressed genes related to growth and development were mainly enriched in the Cell cycle, Endocytosis, TGF-beta signaling pathway, Wnt signaling pathway, FoxO signaling pathway, Insulin signaling pathway, and MAPK signaling pathway. The RNA-sequencing data accuracy was verified by qRT-PCR analysis of the expression levels of selected differentially expressed genes. The abnormal TGF-beta signaling pathway may cause fish weight gain, testis dysplasia, and abnormal spermatogenesis: *smad5*, *smad3a*, *tgfb2*, *tgfbr1b*, *gsdf*, and *amh* were significantly down-regulated. These findings indicated that antisense RNA technology has strong application prospects and can specifically knock down *amh* in Nile tilapia, resulting in an abnormal TGF-beta signaling pathway, inhibiting testis development and inducing weight gain.

**Keywords:** Nile tilapia (*Oreochromis niloticus*); antisense RNA; Anti-Müllerian hormone mRNA inhibition; RNA sequencing; TGF-beta signaling pathway; regulatory mechanism

## 1. Introduction

Antisense RNA can complement or hybridize with protein-coding messenger RNA (mRNA) [1] and specifically block the expression of target genes by interfering with mRNA translation [2]. Antisense RNA technology has been applied to many aspects of synthetic biology. Tomizawa et al. [3] first used this technology to inhibit the production of enterobacter ColE1 plasmid by *Escherichia coli* to improve plant biotype traits [4]. In animal pathology, most experiments have been performed at the cellular level; few in vivo experiments have been performed [5,6]. Uzbekova et al. [7] constructed gonadotropin-releasing hormone (GnRH)-inhibited rainbow trout (*Oncorhynchus mykiss*) by microinjecting a linear DNA fragment into fresh fertilized eggs, and the GnRH levels in the brain of the experimental fish were significantly inhibited and the inhibition was assumed to be permanent. Instead

of microinjection transfection, our laboratory introduced the designed antisense RNA sequence into the ovum through the micropyle to perform a transcriptional knocked-down test of steroidogenic factor 1 (*sf-1*) in Nile tilapia (*Oreochromis niloticus*) [8] achieving a positive transfection efficiency of 88.2%. The down-regulation of *sf-1* resulted in weight gain and gonadal dysplasia in both male and female fish. This method is relatively simple to operate, and not only can it greatly improve the ease of transfection [9] but it also has minimal damage to eggs and the stable phenotype of offspring [8], which is conducive to large-scale production. Antisense RNA has also been used to demonstrate a regulatory role of the genome and has been applied to the differential expression and functional analysis of genes [10]. Therefore, antisense RNA technology has a strong application for gene editing in Nile tilapia.

Anti-Müllerian hormone (*amh*) belongs to a transforming growth factor beta (TGF-beta) superfamily of proteins. This glycoprotein in mammals regulates germ-cell development and differentiation [11]. The TGF-beta signaling pathway includes the TGF-beta/Activin/Nodal (TGF-betas) and Bmp/Gdf/Amh (Bmps) subfamilies [12] and regulates multiple biological processes. Because *amh* induces Müellerian degeneration during embryogenesis in male mammals, it is also known as a Müellerian inhibitor [13]. Although teleosts lack Müellerian ducts, they have *amh* homologues [14]. The expression of *amh* occurs mainly in Sertoli cells and ovarian granulosa cells with specificity and is significantly higher in fish testis than ovarian tissues [15]. In male mammals, Amh binds to specific receptors on target cell membranes, mainly those which promote embryonic development [16] and sex determination and differentiation [17]; *amh* is involved in sex determination in Tiger puffer (*Takifugu rubripes*) [18], Nile tilapia [19], Patagonian silverside (*Odontesthes hatcheri*) [20], and Pike (*Esox lucius*) [21]. Although both *amh/amhy* and *amh/amhrII* may play a critical role in sex determination through the suppression of aromatase expression in teleosts, only the mutation of *amhy* in XY fish resulted in male to female sex reversal [19]. While clustered regularly interspaced short palindromic repeat (CRISPR/Cas9) technology has been recently used to knock out *amh* in fish, how *amh* regulates sex determination is not well understood. We used antisense RNA technology to knock down *amh* to explore the effect of this on gonad development and its molecular regulatory mechanism.

Nile tilapia grows fast, reproduce well, and is of considerable global importance to aquaculture [22]. Although sexually dimorphic [23], there is little difference in the growth of males and females prior to maturity. However, after maturing, the male growth rate exceeds by 30% that of the female [24]. We hypothesized that the continued inhibition of male gonad development will reduce energy expenditure toward reproduction and increase that toward growth [8]. Accordingly, we designed antisense RNA sequences to be introduced into the ovum through the micropyle and constructed a knocked-down model of *amh* mRNA. Changes in the gonadosomatic index (GSI), gonad characteristics, serum hormone levels, and tissue structure were examined. We detected the down-regulation of both relative and absolute transcript levels and protein levels of *amh* by qRT-PCR, absolute quantitative PCR, and Western blot analysis and used RNA sequencing to reveal changes in down-stream genes and pathways resulting from knocked-down *amh*. To promote the application of antisense RNA technology in fish gene-editing, we detailed the experimental process and results. The results of this study provide a possible regulatory mechanism for gonad development in male Nile tilapia.

## 2. Materials and Methods

### 2.1. Ethics Statement

The research protocols and design were approved by the Ethics Committee of the Freshwater Fisheries Research Center of Chinese Academy of Fishery Sciences (FFRC, Wuxi, China). Samples were extracted according to the Guide for the Care and Use of Experimental Animals in China.

### 2.2. Construction of Antisense RNA Knocked-Down Model

2.2.1. Experimental Fish

Nile tilapia was sourced from the Freshwater Fisheries Research Center of the Chinese Academy of Fishery Sciences (FFRC). At a temperature of 28 ± 1 °C, female and male fish were placed separately in indoor 450 L recirculating tanks with pH 7.6 ± 0.2, photoperiod 24 h, dissolved oxygen > 6 mg/L, and ammonia nitrogen content < 0.1 mg/L. Fish were fed a puffed-pellet diet (crude protein 28.0%, crude lipid 6.0%) (Ningbo Tech-Bank Co., Ltd., Ningbo, China) at 8:00 and 16:00 daily. The feed amount represented 4% of the fish body weight. Excrement at the bottom of tanks was cleaned by siphoning daily after fish feeding with ~33% of the water replaced every 3 d. One female and one male with fully developed gonads were selected for eggs and semen.

2.2.2. Design Antisense RNA Sequences

Two antisense RNA sequences were designed and inserted to inhibit expression of *amh* (Figure 1). Antisense RNAs were synthesized by Jinweizhi Biotechnology Co., Ltd. (Suzhou, China).

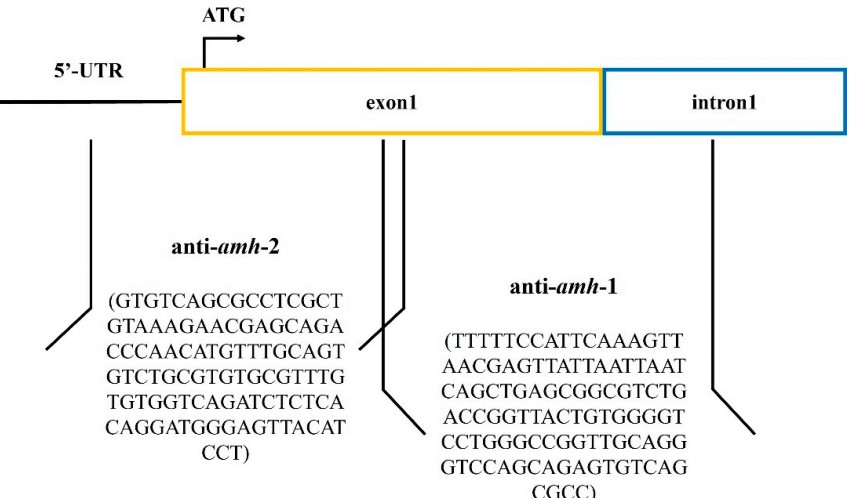

**Figure 1.** Design and action site of two *amh* antisense RNA sequences. Interference sites of the two antisense RNAs (anti-*amh*-1 and anti-*amh*-2).

Antisense RNA sequence 1 of *amh*-1 (anti-*amh*-1): (TTTTTCCATTCAAAGTTAAC-GAGTTATTAATTAATCAGCTGAGCGGCGTCTGACCGGTTACTGTGGGGTCCTGGGCCGGTTGCAGGGTCCAGCAGAGTGTCAGCGCC). To interfere with post-transcriptional *amh* processing, the antisense RNA contained partial first intron and exon sequences.

Antisense RNA sequence 2 of *amh*-2 (anti-*amh*-2): (GTGTCAGCGCCTCGCTGTAAA-GAACGAGCAGACCCAACATGTTTGCAGTGTCTGCGTGTGCGTTTGTGTGGTCAGATCTCTCACAGGATGGGAGTTACATCCT). To interfere with *amh* translation, antisense RNA contained a partial 5-terminal untranslated sequence and a partial sequence of the first exon, including the initiation codon.

2.2.3. PCR Amplification

The two antisense RNA sequences were synthesized and cloned into the pcDNA3.1 expression vector containing the highly expressed cytomegalovirus promoter (Thermo Fisher, Waltham, MA, USA; K482001). A cloning site was located between the Xho I and Xba I restriction sites, and this product was used as the template for subsequent PCR amplification. Forward and reverse primers were designed for template amplification, namely PolyAF1 (GCTTAGGGTTAGGCGTTTTGC) and polyAR1 (TCCCCAATCCTCCCC-CTTGCTG), respectively. The total reaction system was 50 μL, including 2 × Mastermix 25 μL, 2.5 μL of each forward and reverse primer, ultra-pure water 18 μL, and template 2 μL

(antisense RNA fragment carrier). The amplification procedure included pre-denaturation at 95 °C for 2 min and 34 cycles (denaturation at 95 °C for 30 s, annealing at 50 °C for 30 s, and extension at 72 °C for 2 min); this was prolonged at 72 °C for 5 min. In negative control (NC) groups, the blank expression vector was amplified using the PCR program. The 50 μL reaction system comprised 2 × Mastermix 25 μL, primers 5 μL, and ultra-pure water 20 μL.

### 2.2.4. Preparation of Transfection Reagent

PCR amplification products in each group were mixed in a 1:1 ratio, and blank expression vector amplification products (the NC group) or ultra-pure water (the control group) were mixed with lipofectamine 2000 (Thermo Fisher Scientific, Waltham, MA, USA) and sperm preservation fluid (4% sucrose, 3% glycerol, 1% Dimethyl sulfoxide) at a ratio of 4 μL:4 μL:62 μL. The mixture was equilibrated for 30 min at room temperature.

### 2.2.5. Artificial Insemination and Incubation

Mature eggs (250–300) of selected female Nile tilapia were squeezed out manually into each of three clean stainless-steel basins. A 1.5-mL aliquot of buffer (each 1000 mL of ultra-pure water contained 6 g sodium chloride, 1 g glucose, and 0.1 g of each of potassium chloride, calcium chloride, sodium bicarbonate, and sodium dihydrogen phosphate) was added to promote the micropyle opening; 0.8 mL of the transfection reagent was added. The stainless-steel basin was slowly shaken for 15 min to allow antisense RNA fragments to possibly enter eggs through the micropyle. Concomitantly, the corresponding NC group and control group were treated—the NC group with 0.8 mL of transfection reagent containing blank expression vector amplification product, and the control group with 0.8 mL of ultra-pure water. Semen (0.2 mL) from male fish with well-developed gonads (genital papilla ruddy and salient) was gently drawn with a disposable dropper and placed into each receptacle containing eggs. After stirring with goose feathers for 20 s, 2 mL of incubation water was added to complete the artificial insemination process. Fertilized eggs were placed in the incubator tank at 28 °C water temperature and 5.5 L/min water flow speed to ensure fertilized eggs rolled fully. After 96 h, newly hatched fish were collected and counted; the hatching rate was approximately 80%.

### 2.2.6. Experimental Fish Management

Newly hatched larvae were fed (45.0% crude protein, 8.0% fat content) four times daily for 30 d in a small recirculating water system. Males of average body weight 24.35 ± 1.25 g were selected and placed into 12 indoor recirculating water tanks for each of the treatment, control, and NC groups (four repetitions for each treatment group) at a stocking density of 30 fish/m$^3$. Fish were fed extruded pellets (32.0% crude protein, 8.0% fat content) over 30 min (to ensure fish were well fed) at 8:00 and 16:00 daily. After feeding, excrement was cleaned by siphoning from the bottom of tanks. The tank dissolved oxygen level was maintained at >5 mg/L, water temperature at 28 ± 1 °C, pH at 7.6 ± 0.2, and ammonia nitrogen content < 0.1 mg/L. The experiment lasted for 180 d.

### 2.2.7. Detection of Positive Rate of Transfected Experimental Fish

According to the method described by Qiang et al. [9], at 60 days of age, 12 male fish were randomly selected from each treatment. The fish were deeply anesthetized (using MS-222 solution, 200 mg/L), and their gonads were dissected. Genomic DNA was extracted using the MiniBEST Universal Genomic DNA Extraction Kit Ver 5.0 (Takara Bio Inc., Shiga, Japan). The 20 μL PCR reaction system comprised 0.5 μL each upstream and downstream primer (F1: TTTTGCGCTGCTTCGCGATGTAC, R1: TCCCAATCCTCCCCCTTGCTG, concentration 10 mmol/μL), 1 μL genomic DNA, 10 μL Premix Taq (LA Taq Version 2.0, Dalian, China), and 8 μL RNase-free water. The reaction program was as follows: 94 °C for 2 min, then 35 cycles (95 °C 30 s, 50 °C 30 s, 72 °C 2 min), and a final cycle at 72 °C for 5 min.

### 2.3. Sampling

Fish were fasted for 24 h before sampling. Three males were selected from each tank (12 fish per treatment) at 180d. After deep anesthesia with MS-222 solution (200 mg/L), fish were observed, photographed, and weighed. The whole testis tissue was dissected, weighed, and photographed. Three further males were selected from each tank (12 fish per treatment), and blood samples were drawn from their tail veins after deep anesthesia. Blood samples were centrifuged at 4 °C ($1500\times$ *g*, 15 min) and stored at $-20$ °C. Testis tissues were then removed and divided into six sections: five parts into cryovials and frozen in liquid nitrogen for Western blot analysis and RNA extraction (qRT-PCR, absolute quantitative PCR, RNA sequencing, and qRT-PCR validation of RNA-sequencing data), and one part fixed in 4% paraformaldehyde for histology.

### 2.4. Index Determination and Calculation

2.4.1. Determining Growth Performance

GSI was calculated according to the following equation: GSI = [gonad mass (g)/body weight (g)] $\times$ 100%.

2.4.2. Determination of Serum Hormones

Serum levels of estradiol ($E_2$), testosterone (T), follicle-stimulating hormone (FSH), and luteinizing hormone (LH) were measured using ELISA kits (BPE90005, BPE94034, BPE90008, and BPE90009, respectively, Langdon Biotechnology Co., Ltd., Shanghai, China). The antiserum against FSH was raised in rabbits, which was reactive with tilapia, and polyclonal antibody was prepared. The FSH content was determined by a specific and homologous competitive ELISA method (generally according to Mañanós et al. [25]). In this method, a recombinant β-subunit primary antibody against FSH was used to coat a microtiter plate to form a solid-phase antibody. Then, purified FSH was sequentially added to the microtiter plate filled with monoclonal antibodies to combine with the HRP-labeled FSH and form an antibody–antigen enzyme-labeled antibody complex [26]. After washing, the chromogenic solution was added and, under the catalysis of HRP enzyme, a blue substance was produced, which turned yellow when it reacted with acid. The color of the reaction product was positively correlated with the concentration of FSH in the serum. Standard kit products were first diluted with a gradient of 16:8:4:2:1 to prepare standards at different concentrations to construct the standard curve. For each group, three replicates were analyzed. The optical density (OD) of reaction solutions was measured at 450 nm using a Multiskan spectrum microplate spectrophotometer (BioTek Eon, Winooski, VT, USA). The content of FSH in the samples was then determined by comparing the OD of samples to the standard curve. The same procedure was used to determine serum levels of LH. The detection limits of $E_2$, T, LH, and FSH were 25 ng/L, 7.5 nmol/L, 1 ng/mL, and 1.5 ng/mL, respectively.

### 2.5. Hematoxylin-Eosin Staining

Testis tissues were fixed in 4% paraformaldehyde for 4 d, washed several times with phosphate buffered saline, dehydrated with alcohol of different concentrations, clarified with xylene, soaked and embedded with paraffin following Cao et al. [8], after which 5 μm sections were cut on a microtome (Leica RM2235, Leica Microsystems, Wetzlar, Germany). Sections were routinely deparaffinized, hydrated, stained with hematoxylin for 7 min, rinsed with running warm water for 1 min, soaked in 1% alcohol hydrochloride to differentiate for 1 min, and then stained with eosin dye for 5 min. Hematoxylin–eosin-stained sections were dehydrated with gradient alcohol, clarified with xylene, and sealed with neutral resin. Testis tissue was examined by microscope (Leica UB203I, Nussloch, Germany) and photographed. Furthermore, the numbers of germ cells were calculated according to the methods described previously [27].

## 2.6. Western Blot Analysis

The Amh protein and *β-actin* protein of Nile tilapia were used to immunize New Zealand white rabbits to generate polyclonal antibodies following Qiang et al. [9]. A 0.05 g sample of testis tissue was crushed with liquid nitrogen in a mortar, to which 1 mL of ristocetin-induced platelet aggregation buffer (containing 1% of 10 mg/mL phenylmethane-sulfonyl fluoride) was added, and the solution homogenized in a Polytron (PT2500E, KINEMATICA, Lucerne, Switzerland) homogenizer for 1 min at 4 °C. The supernatant containing protein was collected after centrifugation at 12,000× *g* at 4 °C for 15 min, and protein concentration was measured using a BCA protein determination kit (Sigma-Aldrich Inc., St. Louis, MO, USA). Each sample was adjusted to a final protein concentration of 2 µg/µL; 20 µg of total protein was taken for SDS polyacrylamide gel electrophoresis (SDS-PAGE), and 6 × SDS protein loading buffer was added. Proteins were denatured by heating at 100 °C for 10 min, separated by SDS-PAGE, and then transferred to polyvinylidene fluoride membranes using a wet-transfer method. Membranes were sealed in 5% (*w/v*) skim milk powder for 3 h, washed with Tris-buffered saline with Tween, and incubated with primary target gene antibodies overnight at 4 °C. The next day, membranes were washed with Tris-buffered saline with Tween and incubated with the corresponding secondary antibody: rabbit IgG (Cell Signaling Technology Inc., Beverly, MA, USA) for 1 h at room temperature. Color was developed using an ECL Plus Western blot system kit (Amersham Biosciences Inc., Piscataway, NJ, USA) using *β-actin* as an internal reference protein.

## 2.7. RNA Extraction and Reverse Transcription

Total RNA from testis tissue was extracted using a TRIzol kit (Invitrogen, Thermo Fisher Scientific, Waltham, MA, USA). The quality of total RNA was then controlled using a Nano Drop ND-1000 (Nano Drop, Wilmington, DE, USA). RNA integrity was measured by a Bioanalyzer 2100 (Agilent, Santa Clara, CA, USA); an agarose gel was prepared with 0.3 g agarose + 30 mL 1× TAE. Electrophoresis of the PCR products was performed at 180 V constant pressure to detect RNA integrity. Total RNA satisfied the conditions (concentration > 50 ng/µL, RIN value > 7.0, optical density 260/280 > 1.8, and total RNA > 1 µg) for the down-stream experiment and was then refrigerated at −80 °C until use. cDNA was synthesized following manufacturer instructions using the Prime Script RT Master Mix reverse transcription kit (TaKaRa, Dalian, China) and refrigerated at −20 °C until use.

## 2.8. qRT-PCR

An SYBR Premix Ex Taq kit (TaKaRa, Dalian, China) was used to examine gene transcription levels with *β-actin* as an internal reference. Transcription levels of related genes were detected using qRT-PCR and gene-specific primers (Table 1). The 20 µL reaction system contained 0.6 µL (10 mmol/µL) of each up-stream and down-stream primer, 1 µL cDNA, 10 µL 2 × SYBR Premix Ex Taq II, and 7.8 µL ultra-pure water. The response procedure was 95 °C for 5 min, followed by 40 cycles (95 °C for 15 s, 60 °C for 1 min). At the end of the reaction, the dissolution curve program was 95 °C for 15 s, 60 °C for 15 s, and 95 °C for 15 s. Each reaction was replicated three times.

**Table 1.** Sequences of primers used for qRT-PCR.

| Gene Name | Gene Description | Primer Sequence (5′–3′) |
|---|---|---|
| *cyp19a1a* | cytochrome P450 family 19 subfamily A polypeptide 1a | F: 5′-GCTACAGGATCTCGAAGGGC-3′<br>R: 5′-ACCGAACGGCTGAAAGGTAG-3′ |
| *amh* | Anti-Müllerian hormone | F: 5′-GCTTATCCTCCAGCGAGACC-3′<br>R: 5′-TTGGCTCCCAGTGAAACCTC-3′ |
| *sf-1* | steroidogenic factor 1 | F: 5′-TTTGTCCTTCGGCTCAGTCC-3′<br>R: 5′-CGTGTACCTCGGTGTGTTGA-3′ |

**Table 1.** *Cont.*

| Gene Name | Gene Description | Primer Sequence (5′–3′) |
|---|---|---|
| *smad3a* | smad family member 3a | F: 5′-TGGCTGGACAAGGTGCTTAC-3′<br>R: 5′-TTGTGTAGCCGTTCTCGTCC-3′ |
| *smad5* | smad family member 5 | F: 5′-GGCTGAATACGATGACTCCCC-3′<br>R: 5′-GCCTCACTGGTGCAAGTCT-3′ |
| *gsdf* | gonadal somatic cell derived factor | F: 5′-GAGCAGTGGAACCGAACCTT-3′<br>R: 5′-GAACAACACTTCAGGCTCGC-3′ |
| *tgfb2* | transforming growth factor beta 2 | F: 5′-TGCTGTGTCTCCCAAGACCT-3′<br>R: 5′-CGGCACTTTGACGGTACGTT-3′ |
| *tgfbr1b* | transforming growth factor beta receptor 1b | F: 5′-GACTTGATCCCACGAGACCG-3′<br>R: 5′-GGCCACCGGGTCTTTGTT-3′ |
| *dmrt1* | double sex and mab-3-related transcription factor 1 | F: 5′-CGCAGTACCAGATGCCTCAT-3′<br>R: 5′-CAGGCTAAAGAAGGGTGGCA-3′ |
| *β-actin* |  | F: 5′-CCACACAGTGCCCATCTACGA-3′<br>R: 5′-CCACGCTCTGTCAGGATCTTCA-3′ |

## 2.9. Absolute Quantitative PCR

### 2.9.1. Amplification of *Amh* Fragment

Reverse transcription products were used to prepare the absolute quantitative standard and quantitative PCR amplification reaction. The reverse transcription product was used as the template for PCR amplification. The PCR reaction system included 25 μL 2× Taq PCR Master Mix (including dye), 4 μL cDNA, 0.8 μL (10 mmol/μL) of each up-stream and down-stream primer, and ultra-pure water added to a total volume of 50 μL. The PCR amplification procedure involved pre-denaturation at 94 °C for 5 min, 35 cycles (94 °C denaturation for 45 s, 55 °C annealing for 30 s, 72 °C for 1 min), and a 72 °C extension for 10 min. Product was detected by 1% agarose gel electrophoresis.

### 2.9.2. *Amh* Fragment Cloning and Verification

PCR product was purified using an agarose gel DNA recovery kit. Purified product was ligated with PMD619-T overnight at 16 °C. The ligating system included 4.5 μL of the target fragment, 5 μL of solution I, and 0.5 μL of T vector. Ligands were transformed into *E. coli* DH5α competent cells and coated in LB solid medium containing 100 mg/L ampicillin and cultured overnight at 37 °C. The next day, white positive single colonies were selected and inoculated in LB liquid medium containing ampicillin for 1 h of shaking culture. PCR was used for preliminary identification of the bacteria liquid; positive bacteria were selected for plasmid extraction.

### 2.9.3. Plasmid DNA Extraction

The standard plasmid was constructed with *amh* sequences (primer information as in Table 1). The vector was pcDNA3.1(-), the restriction site EcoRI + BamhI, the inserted fragment size 400 bp, and the total plasmid length 5810 bp. The plasmid we constructed was named BK757 PcDNA3.1(-)-*amh*, ammonia benzyl resistance. The plasmid was extracted from the positively identified bacterial solution using an endotoxin-free plasmid small extraction medium volume kit (DP118). Plasmid quality control was performed using a Nano Drop One nucleic acid analyzer (Nano Drop, Wilmington, DE, USA). The plasmid copy number was calculated according to the calculation, and the plasmid was diluted 10 times. Eight plasmids of dilution ($10^{-1}$ to $10^{-8}$) were selected as templates for qRT-PCR, and standard curves were established (Table S1 and Figure S2). The qRT-PCR system included 2 μL cDNA, 0.4 μL (10 μmol/L) of each up-stream and down-stream primer, 10 μL SYBR qPCR Mix, and ultra-pure water to make a final volume of 20 μL. The PCR amplification procedure involved pre-deformation at 95 °C for 30 s, and 40 cycles of reaction

(95 °C for 10 s, 60 °C for 30 s); the dissolution curve program was 95 °C for 15 s, 60 °C for 1 min, and 95 °C for 15 s. A standard curve was drawn using the log value of the copy number as the abscissa and the cycle number as the ordinate. The qRT-PCR reaction conditions and reaction system are as described in Section 2.8. Each reaction was replicated three times. Amplification and dissolution curves and the Ct value were automatically generated from the quantitative fluorescence multiplex polymerase chain reaction.

The calculation of the copy number in 1 ng standard was as follows: $(6.02 \times 10^{23}) \times (1\ \text{ng}/\mu\text{L} \times 10^{-9})/(\text{DNA length} \times 660) = \text{copies}/\mu\text{L}$.

### 2.10. RNA Sequencing

2.10.1. Library Construction and Sequencing

Gonadal RNA of three males extracted from each group (treatment and control) in Section 2.3 was mixed and sequenced with three replicates in each treatment group. Three control (CAMH 1, CAMH 2, CAMH 3) and three treatment (TAMH 1, TAMH 2, TAMH 3) sequence libraries were constructed. Paired-end sequencing was performed using an Illumina Novaseq™ 6000 (LC Bio Technology Co., Ltd., Hangzhou, China) following standard procedures; the instrument produced 150-bp paired-end (PE150) raw reads.

2.10.2. Assemble and Annotate Transcripts

Raw data were obtained in FastQ format. Cutadapt software was used to remove raw data connectors, with low-quality and repeated sequences then removed to obtain clean data in fastq.gz format. Clean data were compared to the genome of Nile tilapia using HISAT2 software to obtain bam files. String Tie software was used for initial gene or transcript assembly, and the initial assembly results of all samples were combined. GffCompare software was used to compare transcripts and reference annotations to obtain final assembly annotation results. BLAST was used to compare valid data with the reference genome.

2.10.3. Identification of Differentially Expressed Genes

Reads per kilobase of exon model per million mapped reads values were used to detect gene transcript abundance [28]. Differentially expressed genes (DEGs) were determined using DESeq2 [29]. DEGs were screened for $|\log2\ \text{fold change}| \geq 1$ and $p < 0.05$ [30]. All DEGs were subjected to Gene Ontology (GO) and Kyoto Encyclopedia of Genes and Genomes (KEGG) enrichment analysis. $p < 0.05$ was considered significantly enriched.

2.10.4. qRT-PCR Validation

After nine DEGs were selected, qRT-PCR was used to verify RNA-sequencing data accuracy. The primer design for DEGs is detailed in Table 1; the experimental method was the same as in Section 2.8.

### 2.11. Statistical Analysis

Growth performance and gene transcription levels in the graphs are presented as mean ± standard error (mean ± SE). Experimental data were analyzed using IBM SPSS Statistics v 22.0. Shapiro–Wilk's and Levene's tests were used to test for normality and homogeneity of variance, respectively. One-way ANOVA and independent samples t-tests were used to compare within-group differences and between-group differences, respectively. Tukey's post hoc test was used to compare treatment groups following one-way ANOVA. Differences were considered significant at $p < 0.05$.

## 3. Results

### 3.1. Determination of Positive Transfection Rate

The gonad tissues of 60-day-old male fish were analyzed using specific primers, and the positive rate of transfection was more than 85%. The sequencing work was completed by Genewiz Biotechnology Co., Ltd., Suzhou, China. The sequencing confirmed that the

two transfected antisense RNA sequences were present in the fish in the treatment group. Each positive sample in the treatment group contained antisense RNA sequences.

### 3.2. Antisense RNA Inhibits Expression of amh mRNA and Protein in Testis

To analyze the inhibitory effect of antisense RNA on *amh*, Western blot analysis, qRT-PCR, and absolute quantitative PCR were used to detect the transcription and protein levels of the target gene in the testis tissues. The Amh protein was detected in the Nile tilapia testis tissue (Figure S1). The Western blot analysis revealed the expression of Amh protein in the treatment groups to be significantly lower than in the control and NC groups. qRT-PCR and absolute quantitative PCR revealed the *amh* mRNA expression levels in the treatment groups to be significantly lower than in the control and NC groups (Figure 2). These results confirmed that the insertion of antisense RNA fragments inhibited the transcription levels of *amh* and Amh protein in the testis tissues and thus down-regulated *amh*.

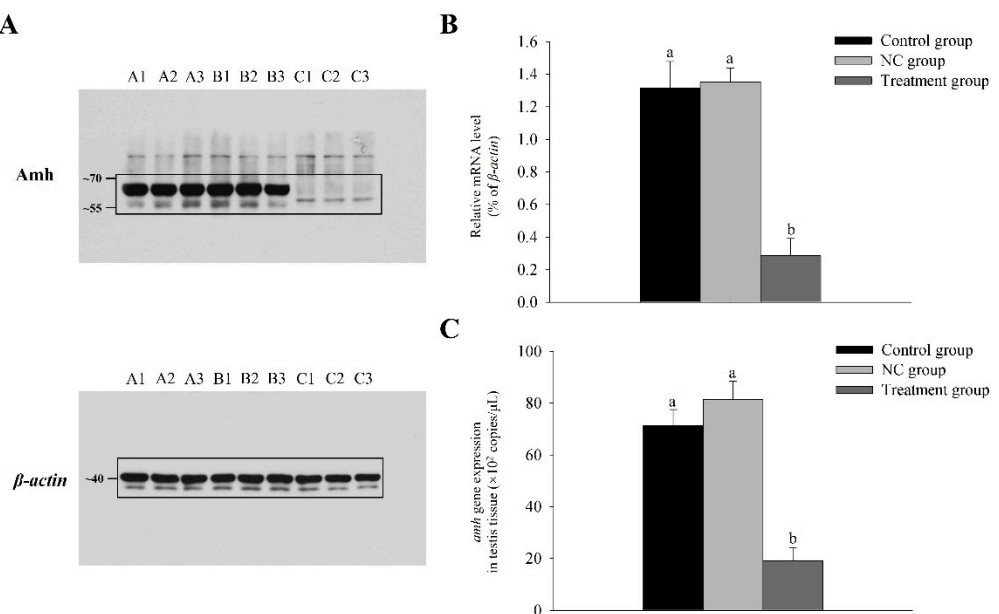

**Figure 2.** Down-regulation of *amh* down-regulated the expression of Amh protein and the mRNA expression level of *amh*. (**A**) Representative images of expression levels of Amh protein in testis tissues of control and treatment groups: A1–A3, control; B1–B3, negative control (NC); C1–C3, treatment. *β-actin* was an internal parameter in each group; (**B**) Transcript levels of *amh* in testis tissues of each group (mean ± SE, *n* = 12 replicates). Identification of *amh* mRNA levels in treatment, control, and negative control (NC) groups as determined by qRT-PCR; (**C**) Absolute quantitative expression levels of *amh* in testis tissues of each group (mean ± SE, *n* = 12 replicates). Identification of *amh* mRNA levels in treatment, control, and negative control (NC) groups as determined by qRT-PCR. Different lowercase letters indicate significant differences ($p < 0.05$).

### 3.3. Knocked-Down Amh Affects Formation of Male Secondary Sexual Characteristics

To observe the changes caused by knocked-down *amh*, we examined and photographed the fish testes. After being cultured for 180 d, the males in the control and NC groups were sexually mature, with obviously red, protruding sexual organs and clear genital openings with a white cylindrical protruding tip. A small quantity of semen was extruded with gentle abdominal compression. The male fish in the treatment groups had a slightly convex white sexual organ, and no semen was extruded with gentle abdominal compression. The males in the treatment groups showed testicular atrophy and short genital papilla (Figure 3).

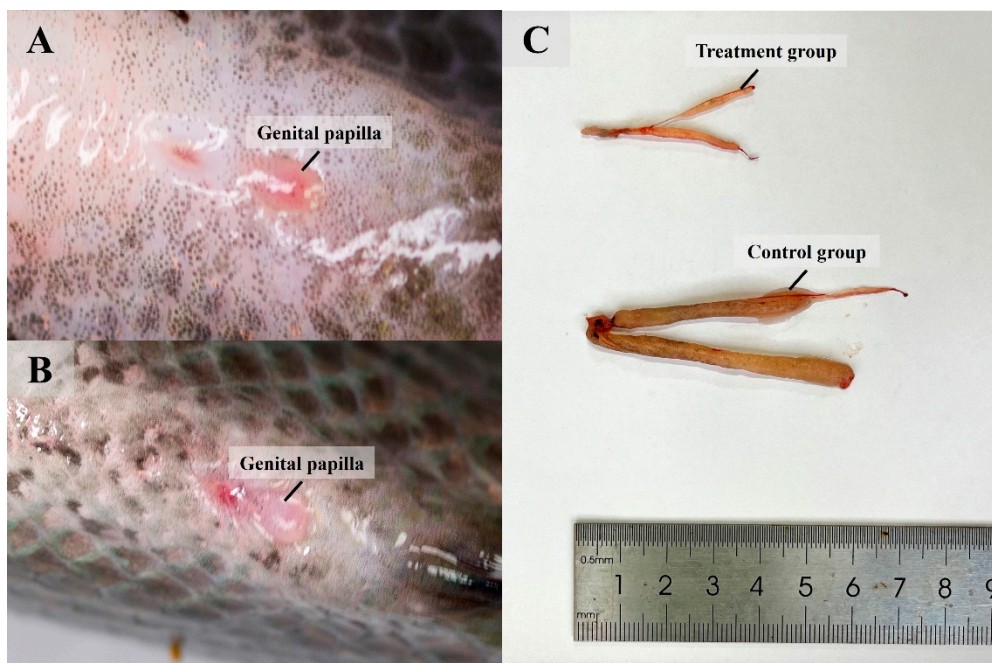

**Figure 3.** Down-regulation of *amh* inhibited development of sex organs and testis tissues in male Nile tilapia. (**A**) Effect of down-regulated *amh* in treatment group; (**B**) male fish in control group; (**C**) Comparison of testis tissue between treatment group and control group.

### 3.4. Knocked-Down Amh Inhibits Testis Development and Maturation

To examine the effects of knocked-down *amh* on testis development, the testes of the fish in each group were weighed and histological sections examined. The testis weight and GSI of the males in treatment groups were significantly lower than those in the control and NC groups ($p < 0.01$), and the body weight was significantly higher than in control and NC groups ($p < 0.05$) (Table 2). Differentiating germ cells was achieved based on hematoxylin–eosin staining. Many sperm (SP) were apparent in the control and NC groups. In contrast, the development of primary sexual characteristics in the treatment group males was delayed, with significant reductions in the numbers of sperm and less dense sperm (Figure 4).

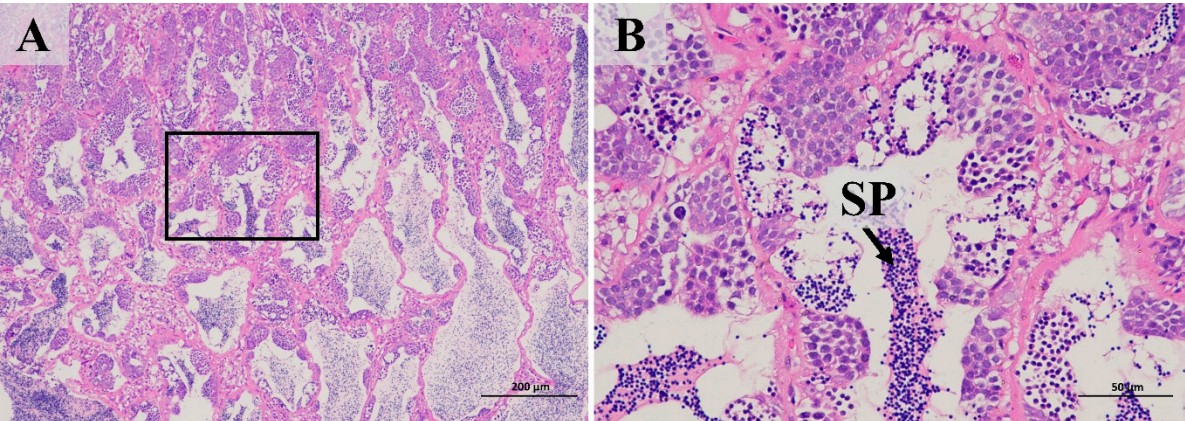

**Figure 4.** *Cont.*

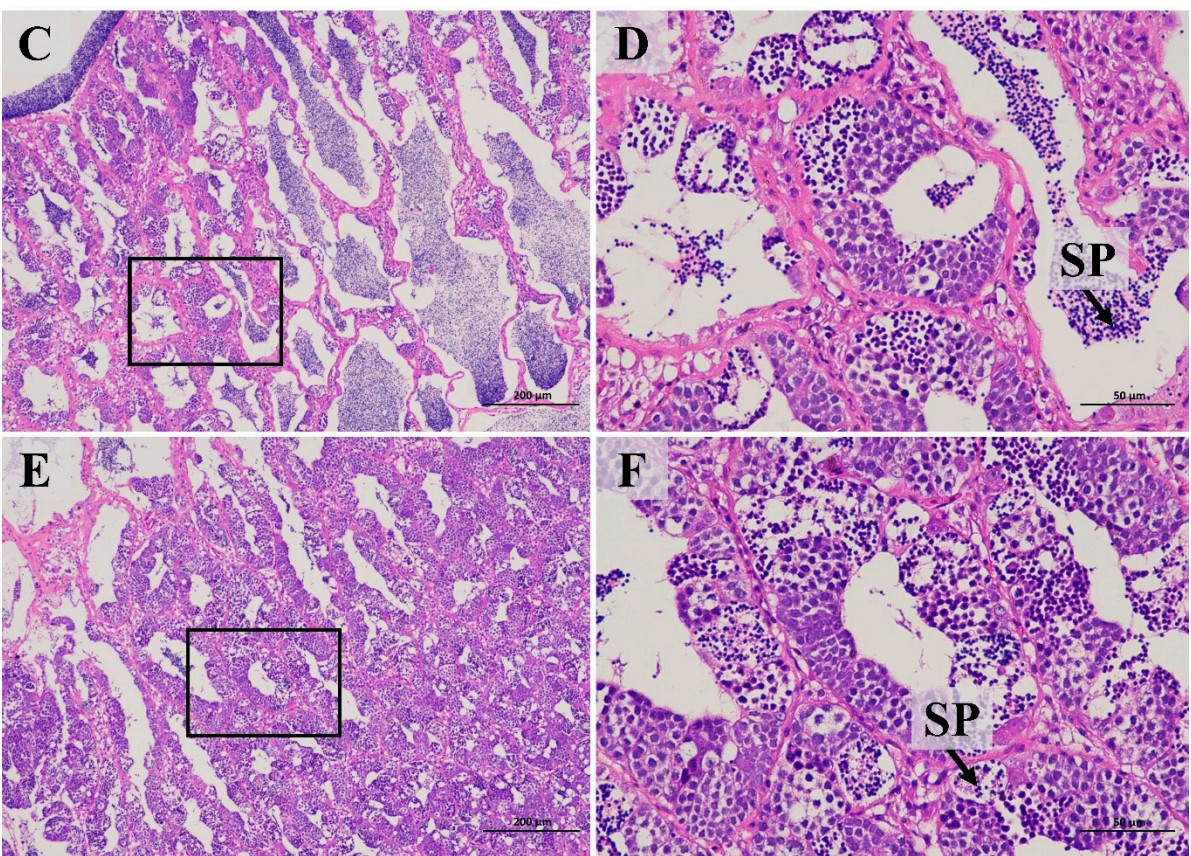

**Figure 4.** Representative sections of testis tissues from male Nile tilapia with knocked-down amh in: (**A**) control group at $100\times$ and (**B**) $400\times$; (**C**) negative control group at $100\times$ and (**D**) $400\times$; and (**E**) treatment group at $100\times$ and (**F**) $400\times$. SP: sperm.

**Table 2.** Fish body weight, gonad weight, and gonadosomatic index (GSI) of each group.

| Measurement | Control Group (n = 12) | Negative Control (NC) Group (n = 12) | Treatment Group (n = 12) |
|---|---|---|---|
| Final body weight (g) | 249.41 [b] ± 18.33 | 242.57 [b] ± 22.14 | 316.76 [a] ± 24.48 |
| Gonadal weight (g) | 2.38 [b] ± 0.63 | 2.41 [b] ± 0.53 | 0.32 [a] ± 0.09 |
| Gonadosomatic index (GSI) | 0.96 [b] ± 0.11 | 1.00 [b] ± 0.13 | 0.10 [a] ± 0.04 |

Data were analyzed by one-way analysis of variance. Differences among three groups were detected using Tukey's multiple comparisons test ($p < 0.05$). Different lowercase letters show significant differences among experimental groups.

### 3.5. Knocked-Down Amh Affects Gonadal Regulation-Related Hormone Levels in Male

Serum $E_2$ levels in the treatment group were significantly higher than in the control and NC groups ($p < 0.05$). Contrarily, the levels of T, LH, and FSH were significantly lower than those of the control and NC groups ($p < 0.05$) (Figure 5).

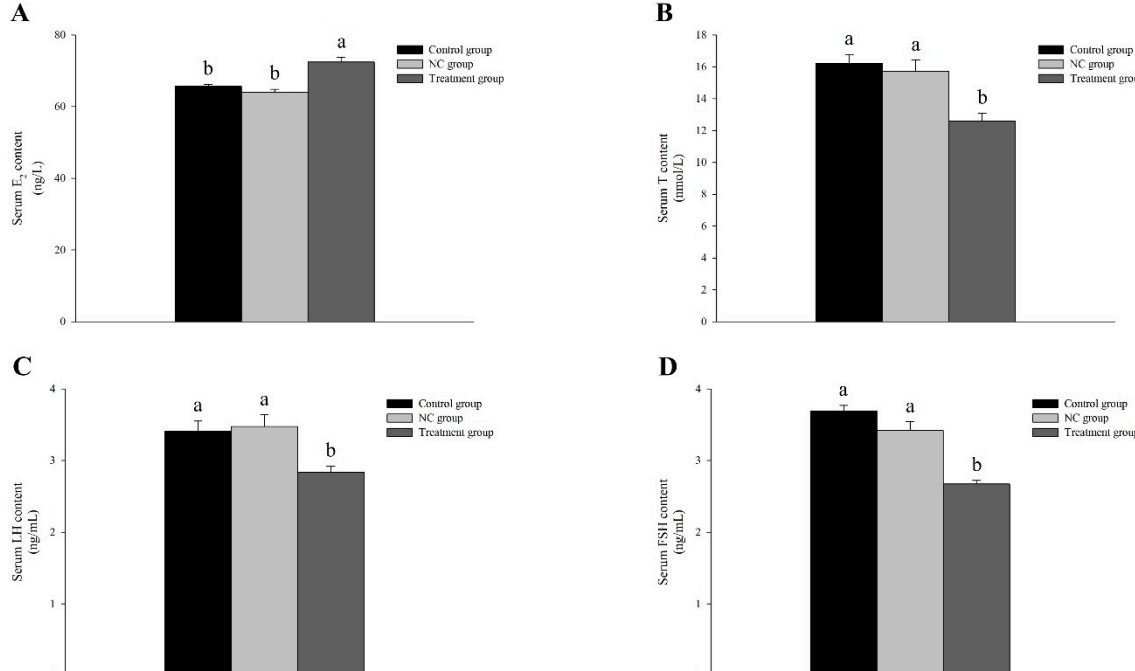

**Figure 5.** Serum hormone contents in Nile tilapia after transfection of vector-encoding antisense RNA (*n* = 12 replicates). (**A**) serum E2 content, (**B**) serum T content, (**C**) serum LH content, (**D**) serum FSH content. Different lowercase letters indicate significant differences (*p* < 0.05).

### 3.6. Transcriptome Analysis Reveals the Effect of Knocked-Down Amh on Testis Development

Based on the resulting genome, we compared the RNA-sequencing results and identified their quality. After removing low-quality sequences, the number of valid reads in each library ranged from 37,271,364–49,543,150. The Q20 values of the six libraries (CAMH 1, CAMH 2, CAMH 3, TAMH 1, TAMH 2, and TAMH 3) ranged from 99.90–99.97%, and the GC content ranged from 46.5–52.5% (Table 3). The results confirmed both the stability of the transcriptome sequencing analysis and the reliability of the sequencing data. In total, 27,088 genes were sequenced, of which 3051 were up-regulated and 21,155 were down-regulated. DEGs were identified on the basis of |log2 fold change| ≥ 1 and *p* < 0.05; 12,048 DEGs were obtained, including 1281 up-regulated and 10,767 down-regulated genes (Figure 6A).

**Table 3.** Overview of RNA-sequencing data and quality filtering.

| Sample | Raw Reads | Valid Reads | Valid Bases (G) | Valid Ratio (Reads) | Q20 (%) | Q30 (%) | GC Content (%) |
|---|---|---|---|---|---|---|---|
| CAMH 1 | 51,507,620 | 49,543,150 | 7.43 | 96.19 | 99.90 | 97.71 | 48.0 |
| CAMH 2 | 43,773,786 | 42,042,262 | 6.31 | 96.04 | 99.91 | 97.71 | 48.0 |
| CAMH 3 | 41,260,474 | 39,431,276 | 5.91 | 95.57 | 99.97 | 98.20 | 48.0 |
| TAMH 1 | 49,581,146 | 43,477,524 | 6.52 | 87.69 | 99.91 | 97.75 | 52.5 |
| TAMH 2 | 47,641,762 | 44,245,380 | 6.64 | 92.87 | 99.92 | 97.82 | 51.0 |
| TAMH 3 | 42,748,522 | 37,271,364 | 5.59 | 87.19 | 99.97 | 98.01 | 46.5 |

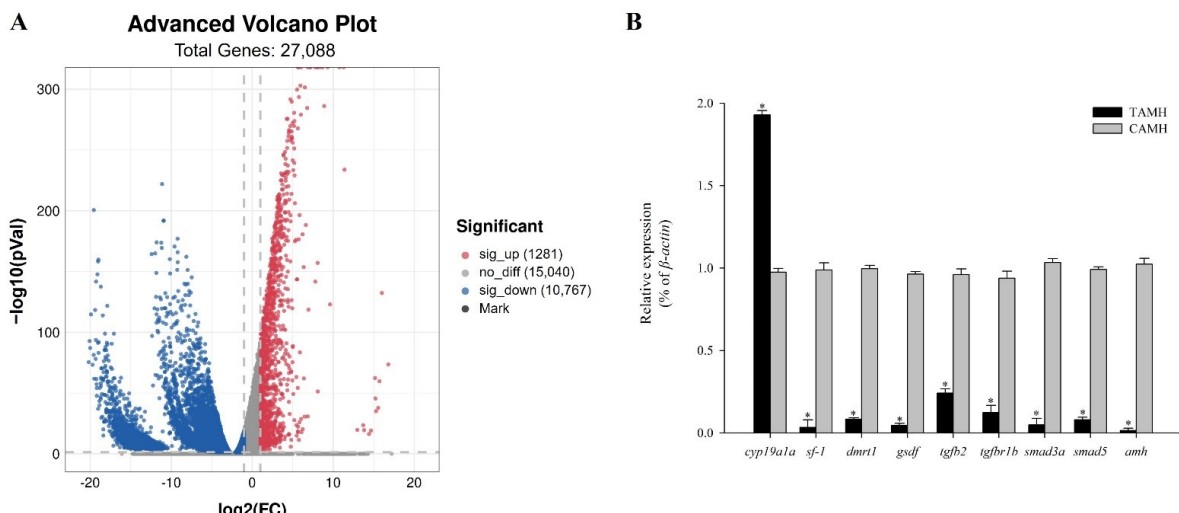

**Figure 6.** (**A**) Volcano plot of differentially expressed genes (DEGs) in Nile tilapia under knocked-down *amh* (treatment groups vs. control groups). Dots: blue, down-regulated DEGs in treatment compared with control groups; red, up-regulated DEGs in treatment compared with control groups; gray, genes with no significant difference in expression. (**B**) Transcript levels of differentially expressed genes (DEGs) in males based on qRT-PCR analyses (*n* = 9 replicates per group). Nine DEGs were selected for qRT-PCR verification. Asterisk (*) indicates significant difference (*p* < 0.05) between TAMH and CAMH.

### 3.6.1. Functional Annotation by GO and KEGG Analysis

The functional classification and enrichment pathways of selected DEGs were analyzed using tools from the GO and KEGG databases. Because of the large number of DEGs, we set the change threshold to *p* < 0.01. Compared with control groups, DEGs in the treatment groups were significantly enriched in 142 GO items, including 51 biological process, 39 cellular component, and 41 molecular function items. The top three significantly enriched items in biological processes were cell cycle, cell division, and DNA repair; in cellular components, cytoplasm, centrosome, and nucleus; and in molecular function, DNA binding, helicase activity, and microtubule motor activity (Figure 7A). The results indicated that knocked-down *amh* significantly affected the cell metabolism and testis development. KEGG pathways related to the development of primordial gonad cells were studied, with seven significantly enriched signal pathways selected: TGF-beta signaling pathway, cell cycle, endocytosis, Wnt signaling pathway, FoxO signaling pathway, Insulin signaling pathway, and MAPK signaling pathway (Figure 7B).

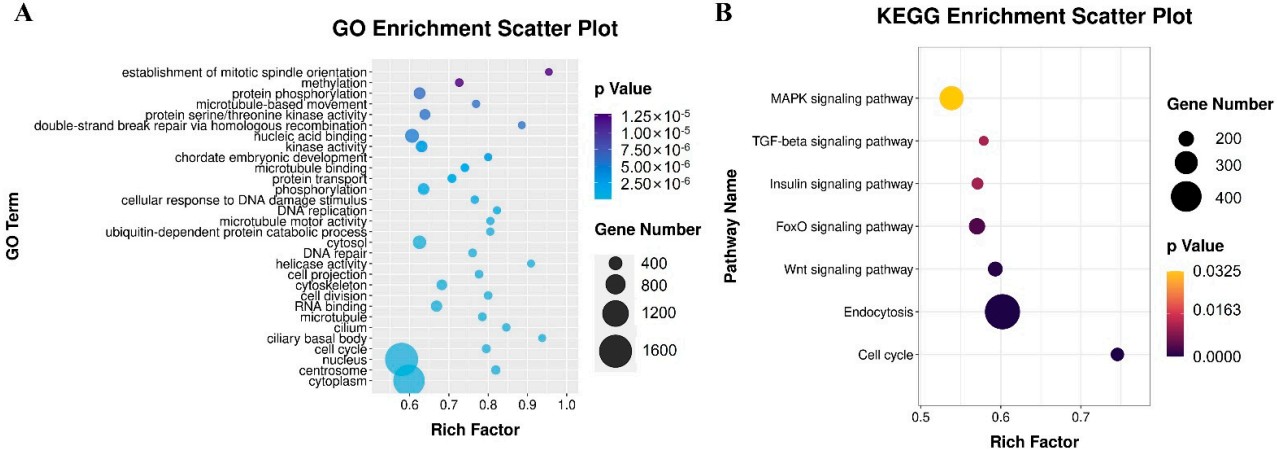

**Figure 7.** Gene Ontology (GO) and Kyoto Encyclopedia of Genes and Genomes (KEGG) enrichment analysis in gonad tissue between *amh*-down-regulated and control groups. (**A**) Top 30 GO catalogs of

differentially expressed genes (DEGs) in testis tissues (treatment groups vs. control groups). Each annotated sequence was assigned at least one GO term in one of biological process, cellular component, or molecular function categories. (**B**) Significantly enriched KEGG pathways associated with cell development and metabolism in Nile tilapia under knocked-down *amh* (treatment groups vs. control groups).

### 3.6.2. TGF-Beta Signaling Pathway

RNA sequencing revealed the TGF-beta signaling pathway to contain multiple DEGs related to embryonic development, fish growth, and the development and regulation of testes. Multiple sex-determining genes in fish were members of this pathway. The KEGG analysis revealed 112 DEGs to be involved in the TGF-beta signaling pathway under knocked-down *amh* (Figure S4). Among the top 20 DEGs, the expression levels of growth differentiation factor 2 (*gdf2*), smad family member 6b (*smad6b*), and bone morphogenetic protein 2b (*bmp2b*) were significantly up-regulated, and those of *amh* and smad family member 1 (*smad1*) were significantly down-regulated (Figure 8).

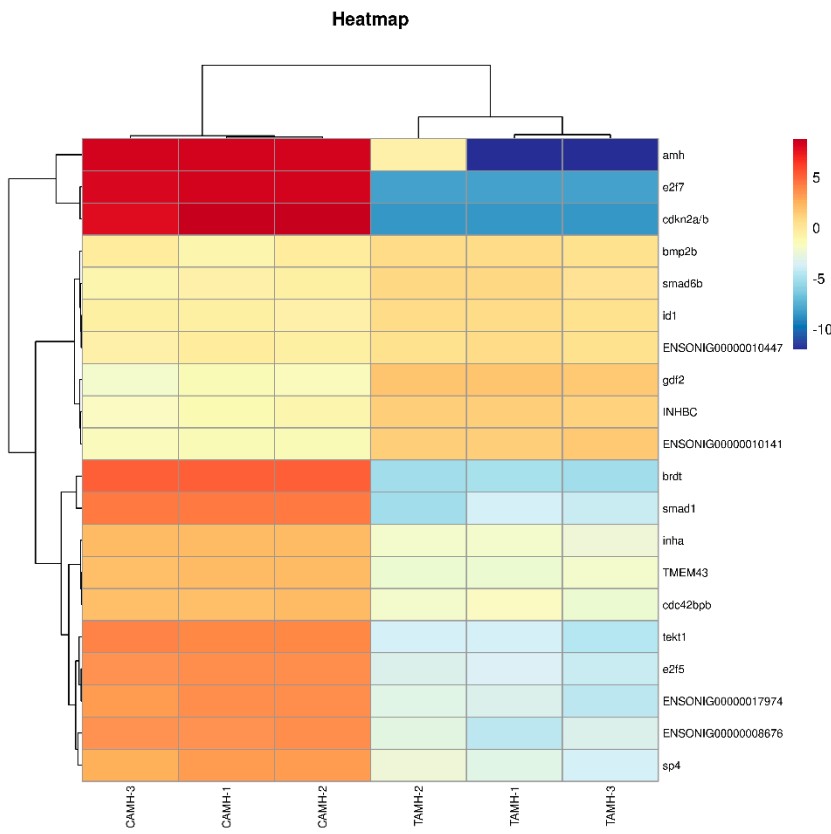

**Figure 8.** Abnormal TGF-beta signaling pathway caused by knocked-down *amh* in Nile tilapia. Heat map of the top 20 differentially expressed genes (DEGs) identified in the TGF-beta signaling pathway. DEGs are defined when the $|\log2$ fold change$| \geq 1$ and $p < 0.05$. Colors are scaled per row. Each square indicates the level of gene expression from highest (red) to lowest (blue).

### 3.6.3. RNA-Sequencing Data Validation

We selected nine DEGs for qRT-PCR analysis. The transcriptional levels of these DEGs were consistent with those in the RNA-sequencing data (coefficient of determination, $R^2 > 0.9$), indicating that RNA-sequencing results were reliable. Compared with the control groups, the transcription levels of smad family member 5 (*smad5*) and smad family member 3a (*smad3a*), transforming growth factor beta 2 (*tgfb2*), transforming growth factor beta receptor 1b (*tgfbr1b*), gonadal somatic cell-derived factor (*gsdf*), double sex and mab-

3-related transcription factor 1 (*dmrt1*), *sf-1*, and *amh* were significantly down-regulated ($p < 0.05$), and the transcription level of cytochrome P450 family 19 subfamily A polypeptide 1a (*cyp19a1a*) was significantly down-regulated ($p < 0.05$) (Figure 6B).

## 4. Discussion

CRISPR/Cas9 [31], which relies on Cas9 and sgRNA to achieve "precise" cutting, has recently replaced zinc finger nucleases (ZFNs) [32] and transcription activator-like effector nucleases (TALENs) [33] to become more widely used in genome editing. CRISPR/Cas9 has enabled great strides in technology and applicability and efficient and accurate biological breeding [34]. However, the application of zebrafish (*Brachydanio rerio var*) [35] and medaka (*Oryzias latipes*) [36] is relatively immature, and there are still many limitations in the application of other aquatic organisms. The reason lies in the low survival rate of its embryos after injection and the possibility of being off target when using it for gene editing, resulting in the mutation of adjacent genes [37]. We introduced antisense RNA into eggs using molecular biology to achieve effective and accurate targeted intervention and regulation of gene expression, which was present in all life stages of the F1 generation and was also detected in the F2 generation. However, the antisense RNA technique was the most effective in inhibiting gene expression in the F1 generation, and gene deletion may have occurred in the F2 generation. Further optimization of the promoter and plasmid was performed to ensure that the antisense RNA could be stably inherited and expressed in future generations. In addition, future research should explore how the transfection sequence enters the egg through the micropyle. Not only is this method relatively simple but egg damage is also minimal, and the progeny have a stable phenotype [8]. We effectively developed an antisense RNA technique to construct a knocked-down *amh* model and reduce transcription and protein expression levels.

### 4.1. Knocked-Down Amh Leads to Increase in Body Weight and Decrease in Male GSI

GSI in teleosts is related to and is an important indicator of sexual maturity [38]. We reported male fish in knocked-down *amh* groups to have abnormal secondary sex characteristics, which may be related to androgen deficiency, including abnormal testosterone production and abnormal spermatogenesis, thus leading to a significant decrease in gonad weight and GSI. *Amh* mutation can also cause gonad dysplasia and dysfunction in zebrafish [39]. Antisense RNA technology produced similar results and was applicable to inhibit target gene expression. The *amh*-deficient males were also heavier than the control group males. The testis weight in the treatment groups decreased significantly, possibly because of the decreased reproductive energy expenditure caused by weight gain [8].

### 4.2. Knocked-Down Amh Inhibits Testis Development

The interruption of spermatogenesis has been described in terms of being delayed or stopped, or in terms of sperm motility being reduced or lost [40]. *Amh* may exert an inhibitory role in the progression of common carp (*Cyprinus carpio*) spermatogenesis [41]. We reported knocked-down *amh* to cause a decrease in sperm production and to interrupt spermatogenesis, possibly because *amh* function is related to the spermatogonial stages of germ-cell development, especially type A spermatogonia [42]. During sex change in Chinese tongue sole (*Cynoglossus semilaevis*), the expression of *amh* is significantly up-regulated. *Amh* may inhibit the proliferation and differentiation of type A spermatogonial cells and regulate the differentiation of male sex in the reversible sex transition process [43]. *Amh* is essential for normal germ cell proliferation. The loss of both *amh* and *amhy* in XY tilapia resulted in significantly increased proliferation of spermatogonia. In addition, *amhy* can compensate for the function of *amh* in controlling germ cell proliferation in the absence of *amh* [44], which may be responsible for a series of changes in testis tissue in this study.

Sex steroid hormones can indirectly and directly regulate sex differentiation of fish [45,46]. Following gene editing using CRISPR/Cas9 to knock out *amhy* in XY Nile tilapia, the expression of *cyp19a1* was up-regulated and serum E$_2$ levels increased, leading

to male-to-female sex reversal [19]. We reported similar results, with increased serum $E_2$ levels in males with knocked-down *amh*. In fish treated with $E_2$, gonad development is inhibited, characterized by reduced GSI and changes in gonad morphology and histology [47–51]. Milnes et al. [40] reported that exposure to estrogen in male fish can inhibit testis growth or cause testis atrophy because of lesions from testis fibrosis or histological changes. Therefore, combined with our results, we concluded that increased $E_2$ levels led to decrease in GSI, atrophy of testes, and changes in testis histology.

Teleost androgens play vital roles in secondary sexual characteristics and behavior and spermatogenesis [52,53]. *Cyp17a1* is essential for testosterone and 11-ketotestosterone production, which further promotes spermatogenesis and fertility in XY males [54]. Testosterone is perhaps the foremost widely studied natural androgenic hormone for growth improvement in fish. In addition, testosterone is a precursor for the production of 11-ketotestosterone in testes [54]. Testosterone is essential for spermatogenesis and male fertility [55]. In the absence of Testosterone signaling, spermatogenesis is halted during meiosis such that few germ cells develop to the haploid spermatid stage and elongated spermatids are not formed [56]. GnRH, secreted by the hypothalamus, stimulates or inhibits secretion of pituitary gonadotropins (LH and FSH) [57]. FSH and LH regulate spermatogenesis and androgen synthesis and release [58,59]. FSH regulates the proliferation and maturation of germ cells independently and in combination with LH [60]. Testosterone is produced by the Leydig cell in response to stimulation with LH and acts as a paracrine factor that diffuses into the seminiferous tubules [55]. We reported levels of LH and FSH in serum of fish in treatment groups to be significantly decreased, accompanied by abnormal spermatogenesis, which may be because *amh* controls the hypothalamic–pituitary function and regulates fertility, as well as the negative feedback of $E_2$ on the hypothalamic–pituitary axis [53]. Moreover, the knock-down of *amh* did not cause sexual reversal because the decrease in serum T, LH, and FSH levels and increase in $E_2$ levels were minimal, and *amhy* may have compensated for the function of *amh*.

### 4.3. Knocked-Down Amh Affects Sex Maintenance in Males

There are four main molecular regulating levels of sex determination and maintenance in teleosts [61,62]. Combining with our results, the first (1) are genes related to sex determination, such as *amhy*. The knock-down of the *amh* did not affect the sex ratio because the sex-determining gene in Nile tilapia is *amhy*. The overexpression of *amhy* in XX fish resulted in female-to-male sex reversal [19]. The second (2) are the up-stream regulatory genes of sex differentiation such as *sf-1* and *amh*, which indirectly or directly regulate the expression of down-stream sex steroid genes and are the initial genes of testis differentiation. The preliminary effects of *sf-1* and *amh* on sex-inversion regulation have been demonstrated. *Sf-1* expression increased significantly during and after testicular differentiation [63]. The down-regulated transcription and protein expression of *sf-1* in Nile tilapia inhibited gonad development and reduced steroid hormone secretion [8]. During the sex inversion in rice-field eel (*Monopterus albus*), the up-regulation of *amh* is likely necessary for the activation of testis development, and the high *amh* expression level facilitates testis function maintenance [64]. We reported similar results, with knocked-down *amh* leading to the down-regulation of *sf-1* expression, increase in serum $E_2$ levels, and impaired testis development. The third (3) is the midstream regulatory gene *dmrt1*, which regulates and maintains gonad differentiation. If the *dmrt1b* of male medaka is knocked out, the male converts to a female [65]. Feeding *amh* plasmid to undifferentiated gonadal, orange-spotted groupers (*Epinephelus coioides*) promoted the expression of *dmrt1*, spermatogonia proliferation, and testis development [66]. We reported the down-regulation of *amh* to lead to the down-regulation of *dmrt1* expression. Finally, the fourth (4) are down-stream regulatory genes, such as androgen receptor (AR), which respond to changes in steroid hormone level expression. Based on gene-level analysis, the RNA-sequencing results confirmed that the knockdown of *amh* resulted in the down-regulation of *sf-1* and *dmrt1* and the up-regulation of *cyp19a1a*. Because Liu et al. [44] reported the up-regulation of *dmrt1* in XY fish to depend

on the suppression of *cyp19a1a*, we believe that sex maintenance in Nile tilapia involves the co-regulation of multiple genes, possibly involving the joint action of the aforementioned four genes.

### 4.4. Molecular Mechanism of Knocked-Down Amh on Male Gonad Development

The KEGG pathway analysis revealed the enrichment of multiple DEGs in multiple pathways. Endocytosis is a way for cells to obtain the necessary nutrients for growth and development [67]. The Wnt signaling pathway [68–70], MAPK signaling pathway [71], FoxO signaling pathway [72], and Cell cycle [73] are involved in regulating cell growth, proliferation, and differentiation. The insulin signaling pathway contributes to glucose storage and uptake [74]. We reported a similar phenomenon in male Nile tilapia with knocked-down *amh*. Changes in gene expression levels in these pathways may affect growth and development, testis development, and germ-cell proliferation.

Triay et al. [75] support the hypothesis that the *amh* region is not the sex-determining region in some wild Nile tilapia populations. We suggest that Nile tilapia may have a polygenic sex determination system. When a major sex determinant is lost, it can be replaced by a new major gene, which may be involved in sex determination and differentiation pathways [75]. The TGF-beta signaling pathway plays an important role in sex differentiation in fish and is involved in mediating various biological processes. Subfamilies of the TGF-beta superfamily (TGF-betas and Bmps), which are transmembrane receptors via type I and II serine/threonine kinases [76], activate two different down-stream Smad pathways (Smad1/5/8, Smad2/3) [77], thereby regulating the transcription of target genes. Smad proteins are important components of TGF-betas [78] and can transmit signals from cell-surface-binding TGF-beta receptors to the nucleus to regulate gene expression [79,80]. *Amhy* determines the male sex of Nile tilapia by repressing *cyp19a1* expression and $E_2$ production through the Amhr2/Smads signaling pathway, resulting in the elevated expression of *dmrt1*, thus initiating male differentiation [44]. Gonad tissue has an endocrine function and can secrete TGF-beta factor. TGF-beta and its receptors can regulate testosterone content, which in turn regulates gonad development [81]. These results suggest that the down-regulated expression levels of *tgfbr1b* and *tgfb2* in the knocked-down *amh* groups also explain the decreased serum testosterone levels, which together led to gonad dysfunction. Bmps are involved in cell growth, apoptosis, morphogenesis, embryonic development, and organogenesis [82]. *Gsdf* plays an important role in spermatogonial proliferation [83], and its overexpression causes transformation of type XX females into functional males. Conversely, the deletion of *gsdf* in XY males leads to increased estrogen levels and male-to-female sex reversal [84]. We obtained similar results, with the down-regulation of *amh* leading to the down-regulation of *gsdf* expression levels, increase in serum $E_2$ level, and gonadal dysplasia and abnormal spermatogenesis of the fish in the treatment groups.

### 5. Conclusions

We designed and applied new gene-editing techniques in fish, which overcame problems with traditional gene-editing techniques for introducing foreign genes and in the selection of target sites. We proposed a response model for Nile tilapia testis based on knocked-down *amh* (Figure 9). The transcription of *amh* was inhibited during the early fertilization of Nile tilapia, indicating the specificity of this gene-editing technique, with a significant down-regulation of *amh* transcription and protein expression, weight gain, suppression of gonad development, and an extremely significant decrease in GSI. An abnormal TGF-beta signaling pathway may cause fish weight gain, gonadal dysplasia, and abnormal spermatogenesis.

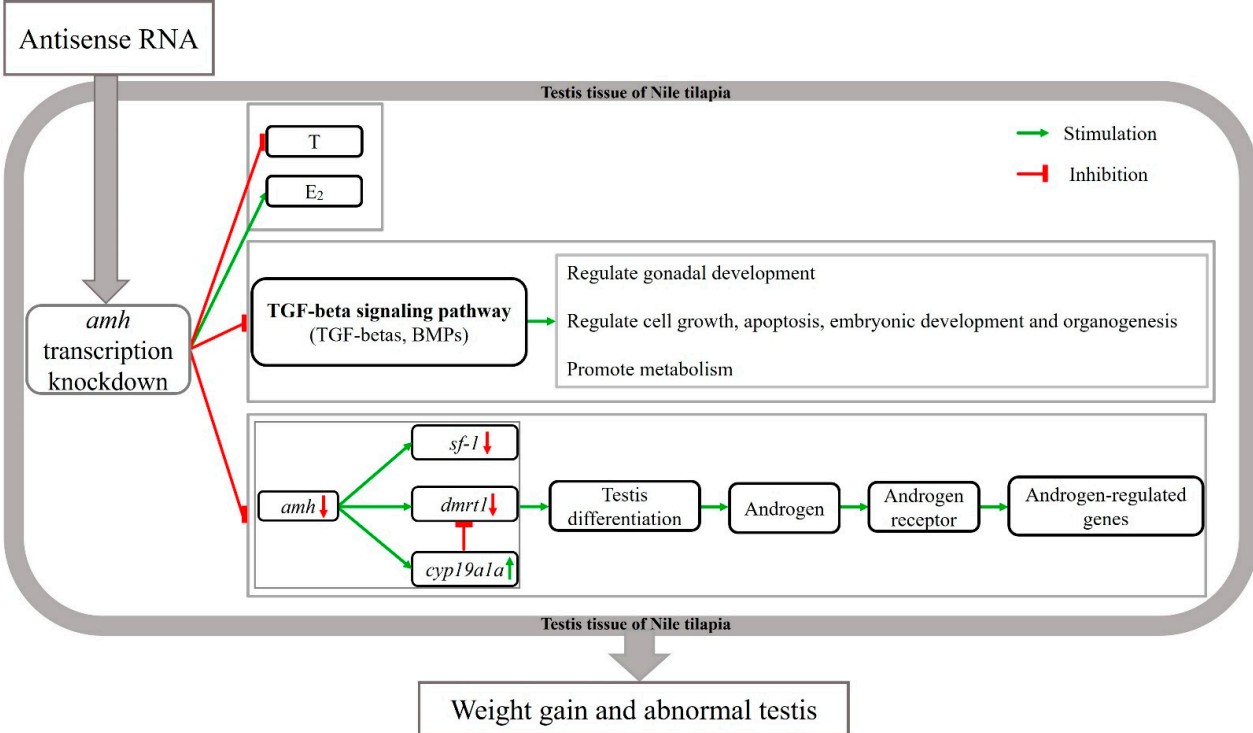

**Figure 9.** Schematic diagram of down-regulated *amh* male Nile tilapia testis development and weight gain regulation.

**Supplementary Materials:** The following supporting information can be downloaded at: https://www.mdpi.com/article/10.3390/fishes7050299/s1, Figure S1: Protein marker map of Anti-Müllerian hormone in testis tissue of Nile tilapia; Figure S2: qRT-PCR standard curve in absolute quantification; Figure S3: The blots for each independent biological replicate used in the analysis; Figure S4: DEGs in TGF-beta signaling pathway. Dots: red, up-regulated DEGs in the TGF-beta signaling pathway (treatment groups vs. control groups); green, down-regulated DEGs (treatment groups vs. control groups). Table S1: Dilution method of standard plasmid for absolute quantification.

**Author Contributions:** The authors thank those persons who gave their time to this research. Conceptualization, Investigation, Methodology, Writing—original draft, Formal analysis, Y.Y.; Formal analysis, Writing—review & editing, Resources, Y.T.; Methodology, Supervision, Validation, Z.C.; Data Curation, Formal analysis, S.L.; Methodology, Supervision, Funding acquisition, P.X.; Conceptualization, Methodology, Supervision, Funding acquisition, Writing—review & editing, J.Q. All authors have read and agreed to the published version of the manuscript.

**Funding:** This study was supported financially by the Central Public-interest Scientific Institution Basal Research Fund, CAFS (No. 2021XT08; 2020TD37); and Central Public-interest Scientific Institution Basal Research Fund and National Natural Science Foundation of China (No. 32002363).

**Institutional Review Board Statement:** The study was conducted according to the guidelines of the Declaration of Helsinki and approved by the Bioethical Committee of the Freshwater Fisheries Research Center (FFRC), Chinese Academy of Fishery Sciences (2013863BCE).

**Data Availability Statement:** Datasets presented in this study can be found in online repositories. Names of repositories and accession numbers are included in the [PRJNA821018 Details | Manage Data | Submission Portal (nih.gov (accessed on 5 April 2022))].

**Conflicts of Interest:** The authors declare no conflict of interest.

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
