# Peer review of "The Effect of Knocked-Down Anti-Müllerian Hormone mRNA on Reproductive Characters of Male Nile Tilapia (Oreochromis niloticus) through Inhibition of the TGF-Beta Signaling Pathway"

_fishes, doi:10.3390/fishes7050299_

Round 1
Reviewer 1 Report
The manuscript employed an amh knock-down model using antisense RNA in male tilapia to investigate downstream effects on sexual development and growth. The study employed a variety of approaches to quantify the molecular responses and was very well conducted. The manuscript was also well written. The authors should consider the following specific comments.
Line 98: please specify the photoperiod used in the study.
L193 2.4.1 Growth performance: please specify how growth (i.e., change in body mass) was determined in addition to GSI. Did the authors also determine Fulton’s condition factor in the tilapia?
L209: detection “limits”
Section 2.4.2: in these in-house ELISAs developed for hormone analyses, additional QA/QC details are essential to assure the readers of their accuracy and precision. What were the intraassay and interassay coefficients of variation for each hormone ELISA? Was parallelism achieved between the standard curves and serial dilutions of unknowns? What are the cross-reactivities of antibodies used to closely related hormones?
L335: please state that Tukey’s posthoc test was used to compare treatment groups following one-way ANOVA.
Fig 3 and Table 2: the images of testes in Fig 3A’’ and 3B”” look to have quite similar sizes, certainly not representative of the data shown in Table 2 (2.4g vs 0.32g). Why were the data for gonad mass and GSI not analysed statistically in Table 2? Please clarify these important points that will cause the readers to question your results.
L521: in teleost fishes 11-ketotestosterone is an important androgen that has key roles in reproduction; perhaps even greater that T due to its greater potency for the androgen receptor. The authors should also include 11KT in their discussion of the role of androgens and estrogens in teleost reproduction
Reviewer 2 Report
The article is very interesting and present new findings in applying of antisense RNA technology in Nile tilapia in order to inhibit testis development and to improving weight gain in this species. Other findings presented in the article are new and valuable as well.
The study provides valid data.
Some specific comments:
Line 42: Tomizawa et al. [3] instead of Tomizawa, Itoh (3)
Line 43: Escherichia coli might be Italic
Line 45: Uzbecova et al. [7] instead of Uzbekova, Chyb (7)
Line 104: please mention the daily amount of feed (in % from BW).
Line 105: How many females and males?
Line 172: Qiang et al. [9] instead of Qiang, Cao (9)
Line 215, 225: : Cao et al. [8] instead of Cao, Qiang (8)
Line 294: (Table S1) ???
Line 517: Milnes et al. [39] instead of Milnes, Bermudez (39)
Line 558: Liu et al. [43] stead of Liu, Dai (43)
Line 571: Triay et al. [43] instead of Triay, Courcelle (74)
Reviewer 3 Report
The manuscript “The effect of knocked-down anti-Müllerian hormone mRNA on
reproductive characters of male Nile tilapia (Oreochromis niloticus) through
inhibition of the TGF-beta signaling pathway” by Yan et al. demonstrates an interesting approach for studying long-term knockdown effects of a gene. However, while interesting and promising, there are a few flaws in how the procedure is presented and interpreted.
The manuscript is generally poorly written, especially the Discussion, with brief non-articulated ideas. It is not only due to the language.
To understand how exactly this technology works, I also read the group’s previous publication describing the same work done on female tilapia as well as some of the cited papers, which led me to ask the following questions:
1. The presence of the plasmid in the gonadal tissue was examined in two months old males. However, if I understand correctly, the detailed experiments were performed on 180 days old fish (you need to clearly specify that in the Material and Methods section). It is unlikely that the plasmid is retained for 6 month and therefore, I tend to believe that the plasmid was incorporated in the genome resulting in a transgenic line. I wonder whether the authors tested germ-line transmission. I believe that this point should be addressed in the discussion.
2. What is the basis for the claim that the plasmid penetrates the egg through the fertilization pore? The use of lipofectamine suggests that the transfection might occur through liposomes crossing the membrane. Please explain.
3. Section 2.2.2. Please explain whether these sequences specifically target the amh on chromosome x or amhy or both. Whether the iRNAs are specific has a significant implication on the interpretation of the results.
4. Another concern is the section describing the hormone measurements by ELISA’s and section 2.4.2. needs to be re-written by addressing the following:
· With regard to LH and FSH Elisa, the description of the procedure is different from the procedure in the cited reference Aizen et al. where the plates were coated with the recombinant beta subunit (and not with antibodies conjugated to horseradish peroxidase- (replace peroxide with peroxidase in line 203)). If the Elisa was performed or given to you by the Levavi-Sivan group, please mention it. If you followed the same procedure, there should be a demonstration of the specificity of the antibodies and quality of the assays.
· I don’t think that the E2 and T Elisa’s are similar to LH and FSH Elisa’s. The concept of small molecules ELISA is usually based on using a tracer. In that case the plates are indeed coated with secondary antibody. Please revisit and correct.
· I don’t understand why GnRH was measured in the serum. GnRH is a neuropeptide and in fish its neurons innervate the pituitary. This means that GnRH is, if at all, minimally secreted to the blood. In addition, the levels presented in Fig.5D are too high to be true in any standard. Not to mention that there is no information if GnRH peptide was measured and what GnRH form (1,2 or 3). This should be clearly specified. If the peptide was measured, a full description of the procedure, including the source of the antibody should be included. Overall however, I don’t see the point of including it in the manuscript and I recommend removing it.
· In line 210: replace lines with limits.
5. Section 4.1: There is no mention of growth rate according to the sub-title, only GSI is mentioned. The entire paragraph is poorly written and structured. Why is amhy KO described here? What is the difference between testicular dysplasia and delayed spermatogenesis?
6. Section 4.3 should also discuss the differences in the roles of amh and amhy genes assuming that only amh was KD. If not, please discuss the suggested option that amhy compensates for the downregulation of amh and how it affects the interpretation of the results. I suggest that a description of the sex differentiation role of amhy versus that of amh will be presented in the Introduction.
7. How the lack of sex-change is explained? Maybe the author should refer to the fact that the decrease in serum T, LH and FSH levels and increase in E2 levels is minimal and link it to amhy.
8. In that regard, how is the big difference in GSI explained? Is it possible that the partial knockdown only delays puberty? Indeed, section 4.3 discusses the downregulation of genes like DMRT1 and upregulation of cyp19a1a. The authors suggest that sex differentiation in Nile tilapia depends on a combination of genes. I argue that the knockdown effect is minimal and insufficient to exert the required changes for sex change or as the author suggested, amhy compensates.
9. Figure 2A. Please add a size marker and indicate whether it is the expected band size. In fact a duplicate figure is found in the supplementary material. Why is there a duplication? It would be better to present the one figure with the kDa marker in the manuscript.
10. Line 20: replace ‘Degenerate testis development’ with ‘impaired testis development’.
11. Line 34: replace ‘improving’ with ‘accelerating’ or ‘inducing’.
12. Line 80: replace ‘posit’ with ‘hypothesized’.
13. Line 85: should be of both relative and absolute transcript levels
14. Section 2.6 and figure 2A: What antibody was used for the control GAPDH or b-actin?
Please provide information about the antibodies used for the WB. Is anti-amh specific?
15. Line 479: mature should immature .
16. Line 481: ‘missing’ should be replaced with ‘off’.
17. Line 490: by “abnormal sex organ cells” the author means “abnormal secondary sex characteristics”?
18. Line 488: the use of down- and up- regulation is generally exclusively used to describe gene expression. Therefore, body weight and GSI increase or decrease. Please correct accordingly. Line 501- replace “down-regulation of sperm” with “decrease in sperm production”.
19. There is no reference in the manuscript to the supplementary material.
20. Figure 9: GnRH, Lh and Fsh are not expressed within the testis, please correct.
Round 2
